# Microvasculopathy and soft tissue calcification in mice are governed by fetuin-A, magnesium and pyrophosphate

Anne Babler[1☯], Carlo Schmitz[1☯], Andrea Buescher[1], Marietta Herrmann[1,2], Felix Gremse[3], Theo Gorgels[4], Juergen Floege[5], Willi Jahnen-Dechent[1] *

1 Helmholtz Institute for Biomedical Engineering, Biointerface Lab, RWTH Aachen University Hospital, Aachen, Germany, 2 IZKF Research Group Tissue Regeneration in Musculoskeletal Regeneration, Orthopedic Center for Musculoskeletal Research, University of Würzburg, Würzburg, Germany, 3 Helmholtz Institute for Biomedical Engineering, Experimental Molecular Imaging, RWTH Aachen University Hospital, Aachen, Germany, 4 University Eye Clinic Maastricht, Maastricht University Medical Center, Maastricht, The Netherlands, 5 Division of Nephrology and Clinical Immunology, RWTH Aachen University Hospital, Aachen, Germany

☯ These authors contributed equally to this work.
* willi.jahnen@rwth-aachen.de

**Data Availability Statement:** All relevant data are within the manuscript and its Supporting Information files.

## Abstract

Calcifications can disrupt organ function in the cardiovascular system and the kidney, and are particularly common in patients with chronic kidney disease (CKD). Fetuin-A deficient mice maintained against the genetic background DBA/2 exhibit particularly severe soft tissue calcifications, while fetuin-A deficient C57BL/6 mice remain healthy. We employed molecular genetic analysis to identify risk factors of calcification in fetuin-A deficient mice. We sought to identify pharmaceutical therapeutic targets that could be influenced by dietary of parenteral supplementation. We studied the progeny of an intercross of fetuin-A deficient DBA/2 and C57BL/6 mice to identify candidate risk genes involved in calcification. We determined that a hypomorphic mutation of the *Abcc6* gene, a liver ATP transporter supplying systemic pyrophosphate, and failure to regulate the Trpm6 magnesium transporter in kidney were associated with severity of calcification. Calcification prone fetuin-A deficient mice were alternatively treated with parenteral administration of fetuin-A dietary magnesium supplementation, phosphate restriction, or by or parenteral pyrophosphate. All treatments markedly reduced soft tissue calcification, demonstrated by computed tomography, histology and tissue calcium measurement. We show that pathological ectopic calcification in fetuin-A deficient DBA/2 mice is caused by a compound deficiency of three major extracellular and systemic inhibitors of calcification, namely fetuin-A, magnesium, and pyrophosphate. All three of these are individually known to contribute to stabilize protein-mineral complexes and thus inhibit mineral precipitation from extracellular fluid. We show for the first time a compound triple deficiency that can be treated by simple dietary or parenteral supplementation. This is of special importance in patients with advanced CKD, who commonly exhibit reduced serum fetuin-A, magnesium and pyrophosphate levels.

**Funding:** This work was supported by grants awarded to WJD by the IZKF Aachen of the Medical Faculty of RWTH Aachen and by the German Research Foundation (DFG SFB/TRR219-Project C-03).

**Competing interests:** The authors have declared that no competing interests exist.

## Introduction

Pathological calcifications, the gradual deposition of calcium phosphate salts [1] in tissue not physiologically meant to mineralize are frequent and are mostly considered benign. However, in particular in the context of chronic kidney disease (CKD), vascular calcifications have increasingly been recognized as a major contributor to cardiovascular morbidity and mortality independent of traditional risk factors [2].

Cell autonomous signaling and the recruitment of mesenchymal cells promote the progression of late stage calcifications to ossifications [3] mimicking bone formation in soft tissues [4]. In most cases, however, calcifications start in the extracellular matrix by nucleation of calcium phosphate crystals in the absence of mineralization regulators [5], before any osteogenic reprograming of resident or invading mesenchymal cells occurs [6, 7].

Phosphate retention in CKD is a major driver of vascular calcification, endothelial damage, and the cardiovascular morbidity and mortality associated with CKD [5]. A disturbed phosphate homeostasis is closely associated with calcifications and accelerated ageing [8]. Consequently, dietary and blood phosphate reduction are prime targets of renal replacement therapy.

Another risk factor for the development of extraosseous calcifications especially in CKD patients is a reduced level of the serum protein fetuin-A [9]. Lack of fetuin-A allows spontaneous mineral nucleation and growth, and hydroxyapatite crystals are deposited causing cardiovascular calcifications [10] and possibly also calciphylaxis [11].

Additionally, ectopic calcification is prevented by low molecular weight ionic compounds pyrophosphate [12], and magnesium [13], which prevent formation of hydroxyapatite through inhibition of crystal nucleation and stability, respectively. Both have been reported to be reduced in sera of patients with advanced CKD [14, 15].

Several years ago we generated mice with a severe spontaneous soft tissue calcification phenotype [16] that worsens progressively throughout life. The mice are deficient in the liver-derived plasma protein fetuin-A, and are maintained against the genetic background DBA/2, which even in the presence of fetuin-A is prone to dystrophic cardiac calcifications [17, 18]. Fetuin-A deficiency in DBA/2 mice greatly worsens the calcification propensity, and is associated with decreased breeding performance and increased mortality. In the fully developed phenotype at about 1 year of age, severe renal calcinosis causes CKD and secondary hyperparathyroidism [16]. From about 4 months of age onward, the mice have myocardial calcification associated with fibrosis, diastolic dysfunction and increased mortality [19, 20]. In contrast, fetuin-A deficient mice maintained against the genetic background C57BL/6 do not calcify spontaneously. However, in a model of CKD induced via nephrectomy and high dietary phosphate diet, these mice developed cardiovascular calcifications [21] mimicking the situation in CKD patients.

Here we studied the calcification in progeny of an intercross of fetuin-A deficient DBA/2 and C57BL/6 mice to identify molecular determinants of their differential calcification that might serve as therapeutic targets. Prompted by the results of our genetic analysis we therapeutically targeted extracellular regulators of mineralization fetuin-A, magnesium phosphate and pyrophosphate and successfully attenuated ectopic calcification.

## Materials and methods

### Animals

Animal experiments were conducted from 2014–2018 in agreement with German animal protection law after approval by LANUV, the state animal welfare committee of the State of

Northrhine-Westfalia (registered trials 87–51.04.2010.A051, 84–02.04.2011.A206, 84–02.04.2014.A452, 84–02.04.2015.A294). Animal health was monitored on a daily basis from birth until the end of experiments and was documented in the institutional animal database as well as in records of laboratory personnel and qualified animal caretakers on animal ID cards. Trained and experienced animal caretakers and investigators have not detected any abnormalities in observed animals regarding their ability to move, access food and water, respiratory changes, body changes.

At different ages as indicated in the text and figures, mice were sacrificed with an overdose of isoflurane and exsanguinated. Animals were perfused with 20 ml ice-cold PBS to rinse blood from the circulation. Wildtype (wt) and fetuin-A deficient (*Ahsg-/-*) mice on DBA/2N (D2) and C57BL/6N (B6) genetic background (ILAR entries D2-*Ahsg*$^{tm1wja}$ and B6-*Ahsg*$^{tm1wja}$ according to ILAR nomenclature) were maintained in a temperature-controlled room on a 12-hour light/dark cycle. Chow (ssniff$^{®}$ R/MH, V1535-0, Soest, Germany) and water were given ad libitum. Mice were kept at the animal facility of RWTH Aachen University Clinics.

For gene segregation and linkage analysis, one D2, *Ahsg-/-* male was mated with one B6, *Ahsg-/-* female. $F_1$ offspring was intercrossed, and 177 $F_2$ offspring were obtained.

## Supplementation therapy

DBA/2 mice deficient for fetuin-A were enrolled in the experiment at three weeks of age. The study included five treatment groups (n≥6 per group) receiving either 0.24 g/kg bodyweight bovine fetuin-A (Sigma, St. Louis, USA) or 0.10 g/kg bodyweight PPi (Sigma) via daily intraperitoneal injections for 3 or 8 weeks, respectively or 1.0% Mg, 0.2% Pi or 0.8% Pi via dietary supplementation for 8 weeks. Control animals received 0.9% NaCl via intraperitoneal injection and normal chow containing 0.2% Mg and 0.4% Pi.

Mice were sacrificed with an overdose of isoflurane, exsanguinated, perfused with ice-cold PBS and organs were harvested. A piece of each of the following tissues (kidney, heart, lung, brown adipose tissue of the neck) was frozen in liquid nitrogen for calcium measurement, another piece was embedded in Tissue-Tek O.C.T. compound and snap frozen for histology.

## High-resolution computed tomography

Mice were anaesthetized with isoflurane and placed in a high-resolution computed tomograph (Tomoscope DUO, CT-Imaging, Erlangen, Germany). The settings of the CT scan were: 65 kV / 0.5 mA, and 720 projections were acquired over 90 seconds. Analysis of the CT scans was performed with the Imalytics Preclinical Software [22]. Full-body acquisitions were obtained to examine in three dimensions the overall state of mineralization in the mice.

## Tissues

At the indicated ages mice were sacrificed with an overdose of isoflurane and exsanguinated. Animals were perfused with 20 ml ice-cold PBS to rinse blood from the vasculature. Calcium content in the tissue (kidney, lung, heart, brown adipose tissue) was determined using a colorimetric o-cresolphthalein based assay (Randox Laboratories, Crumlin, UK). Tissue samples were incubated overnight in 0.6 M HCl followed by homogenization with a Qiagen mixer mill. Homogenates were centrifuged for 10 minutes at 14000 rpm and supernatant was neutralized with ammonium chloride buffer. The subsequent assay was performed according to the manufacturer's protocol.

## Clinical chemistry (Ca, Mg, $P_i$, $PP_i$) and FGF23 serum levels

Blood was clotted and centrifuged at 2000xg for 10 minutes. Serum was harvested and snap-frozen in liquid nitrogen. Standard serum chemistry (Ca, Mg, Pi) was performed by the Institute of Laboratory Animal Science at the University Hospital Aachen. Plasma $PP_i$ was measured in microfiltered plasma as described [23]. Serum FGF-23 was measured with ELISA (Mouse Fibroblast growth factor 23 ELISA Kit, Cusabio, CSB-EL008629MO, Houston, USA) according to manufacturers instructions, with samples diluted 1:2 in sample diluent.

## DNA-analysis for SNP rs32756904 in the gene *Abcc6*

Genomic DNA was isolated from tail tip following established protocols. Using primers flanking the G > A mutation side (`forward, TGGCCCACTCTTGTATCTCC` /`reverse, TTGGGTACCAAGTGACACGA`) a 192-bp PCR fragment was amplified and separated on a 1,5% agarose gel. Amplicons were stained by ethidium bromide, excised with a scalpel under UV light and afterwards purified by QIAquick® Gel Extraction Kit (Qiagen, Hilden, Germany). Sequencing of the PCR products was carried out by Eurofins Genomics (Ebersberg, Germany).

## cDNA synthesis and quantitative real time PCR

Total RNA from the respective organ was isolated with GeneJet RNA Purification Kit (Thermo Fisher Scientific, Waltham, USA). Isolated RNA was treated with DNase I (Roche Diagnostics, Basel, Switzerland) to eliminate genomic DNA. Afterwards total RNA was reverse transcribed (Maxima First Strand cDNA Synthesis Kit, Thermo Fisher Scientific) into cDNA, which served as template both for sequencing (Eurofins Genomics) and for quantitative real-time PCR of three target genes (*Abcc6*, *Trpm6*, *Trpm7*). S1 Table lists the corresponding primer sequences. Primers were previously tested for efficiency. Fold changes were determined using the ΔΔCT method and glyceraldehyde 3-phosphate dehydrogenase *Gapdh* as the reference gene. Maxima SYBR Green/ROX qPCR Master Mix (Thermo Fisher Scientific) was used for qPCR and 20 ng of cDNA were added per reaction. Reaction specificity was determined by the dissociation curve.

By using primers (`forward, CGAGTGTCCTTTGACCGGCT` / `reverse, TGGGC TCTCCTGGGACCAA`) flanking the 5-bp deletion in Abcc6 cDNA of fetuin-A deficient mice on DBA/2N a 144-bp or 139-bp PCR fragment was amplified representing wildtype and mutant alleles, respectively. For the separation of the 5-bp shortened mutant fragment from the wild-type amplicon, high-resolution electrophoresis was performed using a 10% polyacrylamide gel with an acryl-amide/bisacryl- amide-relation of 19:1. Sequencing of PCR products was carried out by Eurofins Genomics.

## Immunoblotting

For Abcc6 Western blot analysis liver samples frozen in liquid nitrogen were powdered with a mortar. Using Lysis Buffer (10 mL/ g organ; 150 mM NaCl, 20 mM Tris base pH 7.5, 1% Triton X-100, 1 mM EDTA, 1xRoche inhibitor cocktail (Roche Diagnostics GmbH)) the proteins were extracted for 30 minutes at 4°C by shaking. Cell fragments were separated by centrifugation (4°C and 20000xg for 15 minutes). The protein concentration of the supernatant was determined using a Pierce BCA Protein Assay Kit (Thermo Fisher Scientific). Protein samples (100 μg/lane) were separated using a gradient gel (NuPAGE® Novex 4–12% Bis-Tris Gel 1.0 mm, Thermo Fisher Scientific) and transferred onto nitrocellulose membrane (Amersham Protran, Sigma Aldrich) by wet electroblotting (Mini Trans-Blot® Cell, Biorad, Hercules,

USA) at 120 V for 2,5 h. The membrane was incubated for 45 minutes at room temperature in blocking solution comprising PBS-T (PBS, 0.05% Tween-20) and 3% nonfat dried milk powder (Applichem, Darmstadt, Germany). Primary antibody (MRP6/M6II-68 monoclonal rat anti-mouse Abcc6 [24]) was diluted 1:1000 in blocking solution containing 1% milk powder and incubated over night at 4˚C. Secondary antibody (horseradish peroxidase coupled rabbit anti-rat IgG, (Dako, Jena, Germany) was diluted 1:2500 in 1% blocking solution and incubated for 1 hour at room temperature. Following antibody incubations membranes were washed three times with PBS-T for 5 minutes. Bound antibody was detected by chemiluminescence in substrate solution (0.1M TRIS/HCl, pH 8.5, 1.25mM 3-aminopthalhydrazide, 0.45mM p-coumaric acid, 0.015% hydrogen peroxide) using a fluorescence imager (Fuji LAS Mini 4000, Fujifilm, Minato, Japan). Abcc6 protein expression was quantified using Image J software (Rasband, http://imagej.nih.gov/ij/).

For serum fetuin-A analysis murine serum was separated by sodium dodecyl sulphate polyacrylamide gel elecrophoresis using mini gels (10% acrylamide, $5 \times 8 \times 0.1$ cm$^3$, BioRad). Protein transfer onto nitrocellulose membrane was carried out by semi-dry electroblotting (Owl HEP-1, Thermo Fisher Scientific) at 6 V for 60 minutes. The membrane was blocked with PBS-T/5% milk powder for 30 minutes at 37˚C. Primary antiserum (AS386 polyclonal rabbit anti-mouse fetuin-A) was diluted 1:2500 in blocking solution and applied for 1 hour at 37˚C. Secondary antibody (horseradish peroxidase coupled swine anti-rabbit IgG, Dako) was diluted 1:5000 in blocking solution and incubated for 1 hour at 37˚C. Following antibody incubations membranes were washed three times with PBS-T for 5 minutes and bound antibody was detected by chemiluminescence as described above.

## Histology

6 μm thick sections were cut and fixed with zinc fixative (BD Biosciences, Heidelberg, Germany). Subsequently, H/E staining or von Kossa staining was performed. The sections were analyzed using a Leica DMRX microscope (Leica Microsystems GmbH, Wetzlar, Germany) and DISKUS software (Carl H. Hilgers, Technisches Büro, Königswinter, Germany).

## Statistics

All statistical analyses were carried out using GraphPad Prism version 8 and are given as mean values ± standard deviations. Student T-test was used for comparison of single treatments versus control group. One-way ANOVA with Tukey multiple comparison test was used to test for differences in multiple experimental groups.

## Results

### Severe ectopic calcification in fetuin-A deficient mice

We performed computed tomography (CT) and calcified lesion volume rendering in wildtype and fetuin-A deficient DBA/2 and C57BL/6 mice. The expected phenotypes were confirmed by serum immunoblots (S1 Fig). Fig 1A–1D show typical CT representations and Fig 1E shows calcified lesion quantification in mice at age 3 months. Fig 1A, 1B, 1D and 1E show that wildtype DBA/2 and C57BL/6 mice as well as fetuin-A deficient C57BL/6 mice had little, if any calcified lesions in their interscapular brown adipose tissue, with total lesion volumes of $0.36 \pm 0.59$ mm$^3$, $0.00 \pm 0.0$ mm$^3$ and $0.02 \pm 0.02$ mm$^3$, respectively. In contrast, DBA/2 fetuin-A deficient mice had more than 100-fold combined calcified lesions of $119 \pm 116$ mm$^3$ in their brown adipose tissue, the skin rubbing against the humerus, the heart and the kidneys (Fig 1C and 1E).

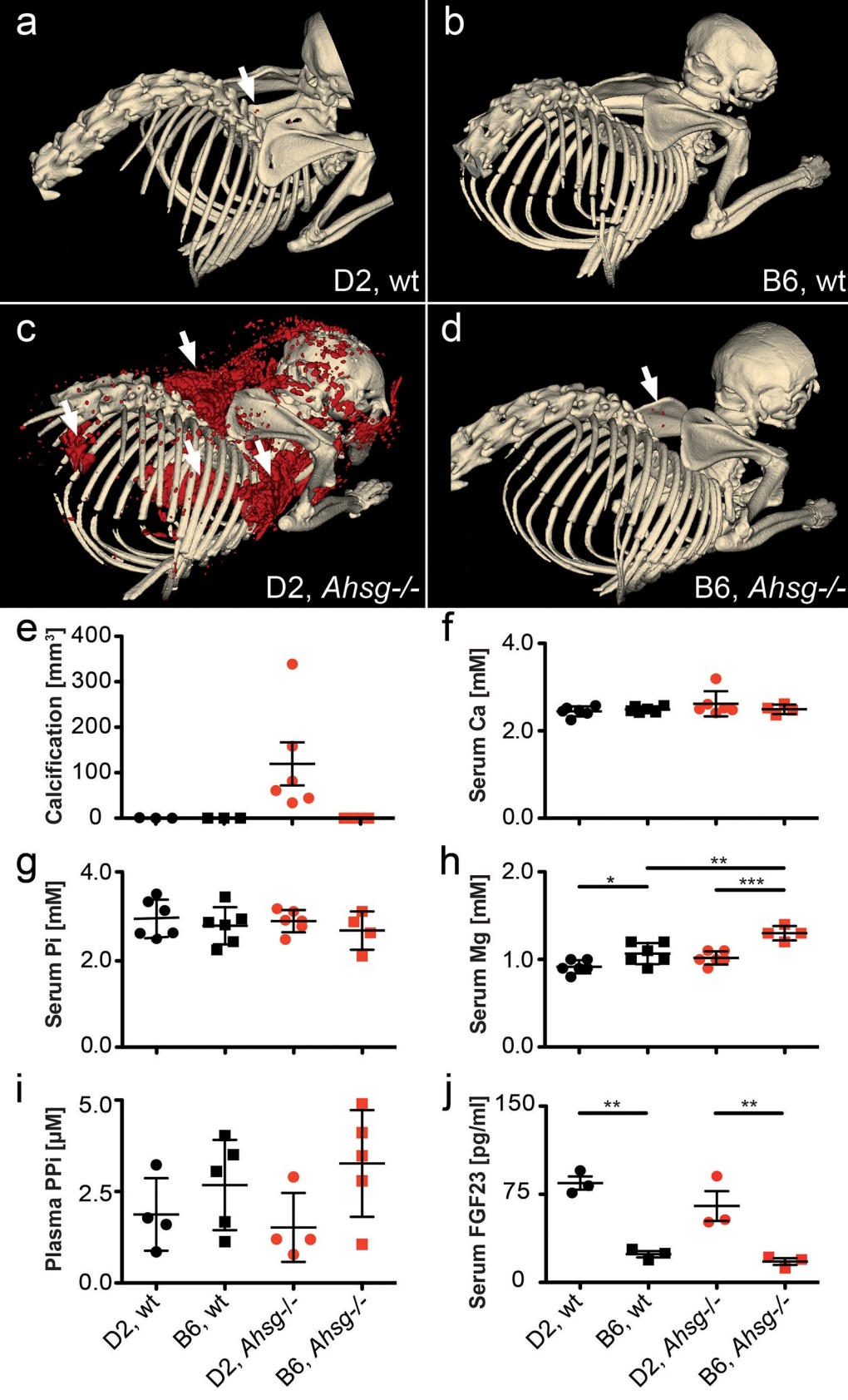

**Fig 1. Extensive soft tissue calcification and hypomagnesemia at age 3 months. a, b,** Wildtype (wt) and **c, d,** fetuin-A-deficient (*Ahsg-/-*) mice maintained against the genetic background DBA/2 (D2) **(a, c)** or C57BL/6 (B6) **(b, d)** were analyzed by computed tomography, and calcified lesions were segmented (red color) and **e,** quantified. Arrows point to small calcified lesions in **a**, the interscapular brown fat tissue of D2,wt and **d**, B6,*Ahsg-/-* mice. **c,** Arrows depict from left to right, massive calcified lesions in the left kidney, the heart, the interscapular brown adipose tissue BAT, and the axillary skin. **f-i,** Serum chemistry of calcification related electrolytes demonstrated **f,** normal serum total calcium, **g,** normal serum phosphate Pi, **h,** hypomagnesemia in D2,wt mice compared to B6,wt mice, and failure to induce serum magnesium in D2, *Ahsg-/-* compared to B6, *Ahsg-/-*, resulting in functional hypomagnesemia, **i,** plasma pyrophosphate, PPi was higher in B6 mice than D2 mice, but differences did not reach statistical significance due to large standard deviation. **j,** serum FGF23 was significantly elevated in D2 compared to B6 mice, regardless of fetuin-A genotype. One-way ANOVA with Tukey multiple comparison test for statistical significance, $^*p<0.05$, $^{**}p<0.01$, $^{***}p<0.001$.

Recently, we showed that ectopic calcification in DBA/2 fetuin-A deficient mice starts in the microvasculature and is associated with premature ageing and cardiac failure, and that cellular osteogenesis was not involved [20]. Thus, we concentrated on extracellular regulators of calcification. Fig 1F and 1G show that serum calcium (Ca) and phosphate (Pi), major ionic drivers of calcification, were similar in DBA/2 and C57BL/6 wt and fetuin-A deficient mice. In contrast, serum magnesium (Mg), a potent inhibitor of calcification, was generally lower in DBA/2 mice confirming previous publications [16, 25] (Fig 1H). C57BL/6 fetuin-A deficient mice, which do not spontaneously calcify, had increased serum magnesium levels compared to their wildtype littermates ($1.30 \pm 0.08$ mM vs. wildtype $1.07 \pm 0.12$ mM, p = 0.005). In contrast fetuin-A deficient DBA/2 mice, which severely calcify, did not display elevated serum magnesium levels compared with wildtype littermates ($1.02 \pm 0.08$ mM vs. wildtype $0.92 \pm 0.08$ mM, p = 0.26). Thus, in addition to the already lower serum Mg, adaptive induction of serum Mg was compromised in DBA/2 mice. Plasma levels of inorganic pyrophosphate (PPi), another systemic calcification inhibitor, were elevated in C57BL/6 mice compared to DBA/2 mice independent of fetuin-A genotype ($2.68 \pm 1.23$ µM vs. $1.87 \pm 0.99$ µM in wildtype mice, p = 0.75 and $3.27 \pm 1.46$ µM vs. $1.52 \pm 0.94$ µM in fetuin-A deficient mice, p = 0.18, Fig 1I). Measured differences were not statistically significant due to large standard deviation, a known complication of measuring PPi in small volumes of plasma. Serum FGF23 levels (Fig 1J) were significantly elevated in DBA/2 mice compared to C57BL/6 mice ($84.5 \pm 9.7$ pg/mL vs. $24.0 \pm 4.7$ pg/mL in wildtype mice, p = 0.002 and $65.1 \pm 22.0$ pg/mL vs. $17.8 \pm 5.0$ pg/mL in fetuin-A deficient mice, p = 0.007) regardless of their fetuin-A genotype suggesting that the DBA/2 mice suffered from mineral imbalance despite apparently normal serum phosphate (Fig 1G). Taken together our results suggest that DBA/2 fetuin-A knockout mice have combined deficiencies of three potent circulating inhibitors of calcification, namely fetuin-A, Mg and PPi.

## Parenteral fetuin-A supplementation attenuates soft tissue calcification

We next asked if supplementation of the circulating inhibitors would attenuate the strong calcification phenotype of fetuin-A deficient DBA/2 mice. First, we injected intraperitoneally (i. p.) 0.24 g/kg bodyweight purified bovine fetuin-A five times weekly for a total of three weeks and then analyzed soft tissues by orthogonal methods histology, tissue chemistry, and X-ray analysis. Fig 2A–2D show that saline injected fetuin-A deficient DBA/2 mice had severe dystrophic calcification in brown adipose tissue with lesions ranging from about 20 µm to several hundred µm in diameter (Fig 2A), few small roundish calcified lesions in kidney and lung (Fig 2B and 2C) and extensive fibrosing calcified lesions in the heart (Fig 2D). Fetuin-A injection i. p. greatly reduced the number and size of calcified lesions (Fig 2E–2H, S2 Fig) as well as the tissue calcium (Fig 2I). The soft tissue calcium content in mice injected i.p. with fetuin-A was reduced in all organs analyzed. Compared to saline injected mice, fetuin-A injected mice had

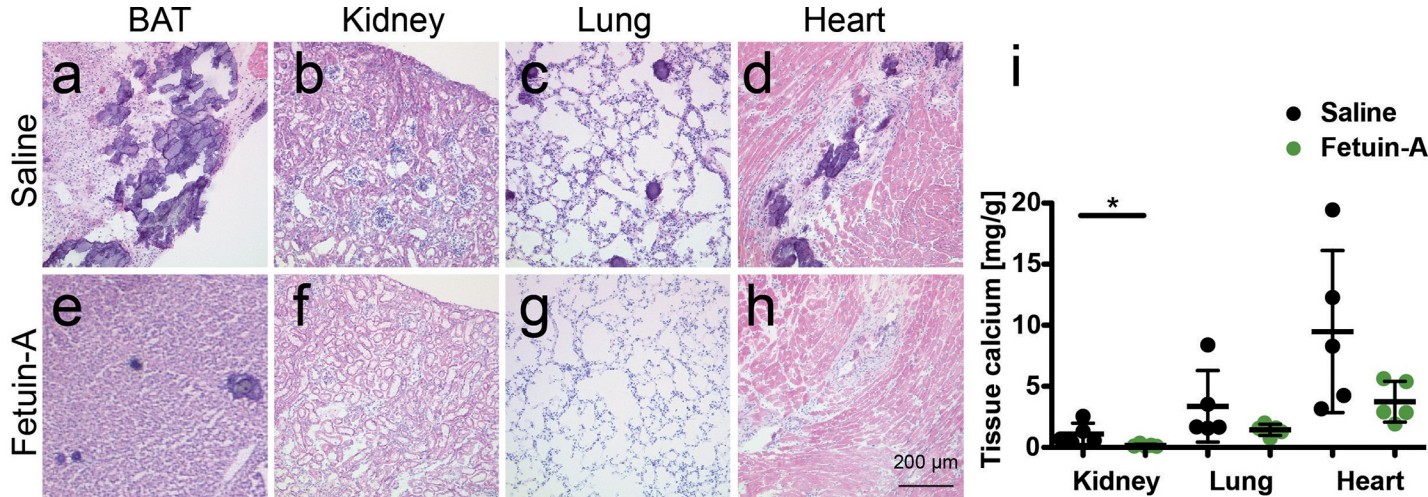

**Fig 2. Parenteral fetuin-A supplementation attenuates soft tissue calcification in D2, *Ahsg*-/- mice. a-d**. Three-week-old mice were injected intraperitoneal (i.p.) five times a week for three weeks, with saline or **e-h,** with 0.24 g/kg bodyweight fetuin-A. Shown are HE stained cryosections at age 6 weeks of **a,e** brown adipose tissue (BAT), **b,f,** kidney, **c,g,** lung, and **d,h** heart. Scale bar indicates 200 µm. Saline treated mice had numerous large calcified lesions in their BAT, occasional lesions in kidney and lung, and extended fibrosing calcified lesions in myocard. In contrast, fetuin-A treated mice developed far less calcified lesions in the same organs. **i,** Tissue calcium content was measured following chemical extraction of kidney, lung and heart of the animals. Treatment with fetuin-A markedly reduced the amount of tissue calcium in all analyzed organs. Student t-test for statistical significance, *p<0.05, **p<0.01, ***p<0.001.

6.9-fold reduced kidney calcium (1.11 ± 0.87 mg calcium/g tissue vs. 0.16 ± 0.11 mg/g, p = 0.043), 2.3-fold reduced lung calcium (3.36 ± 2.93 mg/g vs. 1.44 ± 0.46 mg, p = 0.184), and 2.5-fold reduced myocardial calcium (9.48 ± 6.63 mg/g vs. 3.74 ± 1.67 mg/g, p = 0.097).

## Dietary magnesium supplementation attenuates soft tissue calcification

Quantitative CT volume rendering of calcified lesions showed that dietary Mg supplementation strongly attenuated soft tissue calcification. Similar to the mice shown in Fig 1C and 1E, fetuin-A deficient DBA/2 mice on normal chow (0.2% Mg) developed 101 ± 53 mm$^3$ calcifications within the first 11 weeks postnatally (Fig 3A, 3B and 3E). In contrast, mice fed 1% Mg chow from postnatal week 3 onward developed 7-fold less calcifications i.e. 14 ± 5 mm$^3$ (Fig 3C, 3D and 3E). Chemical tissue calcium analysis and von Kossa histology confirmed that feeding high Mg diet decreased calcifications in all organs analyzed (Fig 3F–3N). Compared to mice on normal chow, mice on high dietary Mg had similarly 10.9-fold reduced BAT calcium (15.46 ± 2.62 mg calcium /g tissue vs. 1.42 ± 0.61 mg/g, p<0.0001), 7.6-fold reduced kidney calcium (1.89 ± 0.65 mg/g vs. 0.25 ± 0.15 mg/g, p = 0.0006), 8-fold reduced lung calcium (2.49 ± 1.10 mg/g vs. 0.31 ± 0.16, p = 0.002), and 5.9-fold reduced myocardial calcium (4.92 ± 1.89 mg/g vs. 0.83 ± 0.61 mg/g, p = 0.002).

To study endogenous Mg metabolism in the mice, we analyzed mRNA expression of two renal and gastrointestinal magnesium transporters Trpm6 and Trpm7. Fig 3O and 3P show that renal Trpm6 mRNA expression in relative units RU was 3.9-fold elevated in fetuin-A-deficient C57BL/6 mice compared to wildtype C57BL/6 mice (5.8 ± 3.0 vs. 1.5 ± 0.4, p = 0.004), while fetuin-A deficient and wildtype DBA/2 mice had similar renal Trpm6 mRNA expression (1.2 ± 0.6 vs. 1.9 ± 1.7, p = 0.88). Trpm6 mRNA expression in gut was similar in all four mouse strains. Fig 3Q and 3R show that Trpm7 mRNA expression likewise was similar in kidney and gut tissue of all mice irrespective of genotype. Collectively, these data indicate that *Trpm6* gene induction mediates the elevated serum magnesium in fetuin-A deficient C57BL/6, but not in DBA/2 mice (Fig 1H).

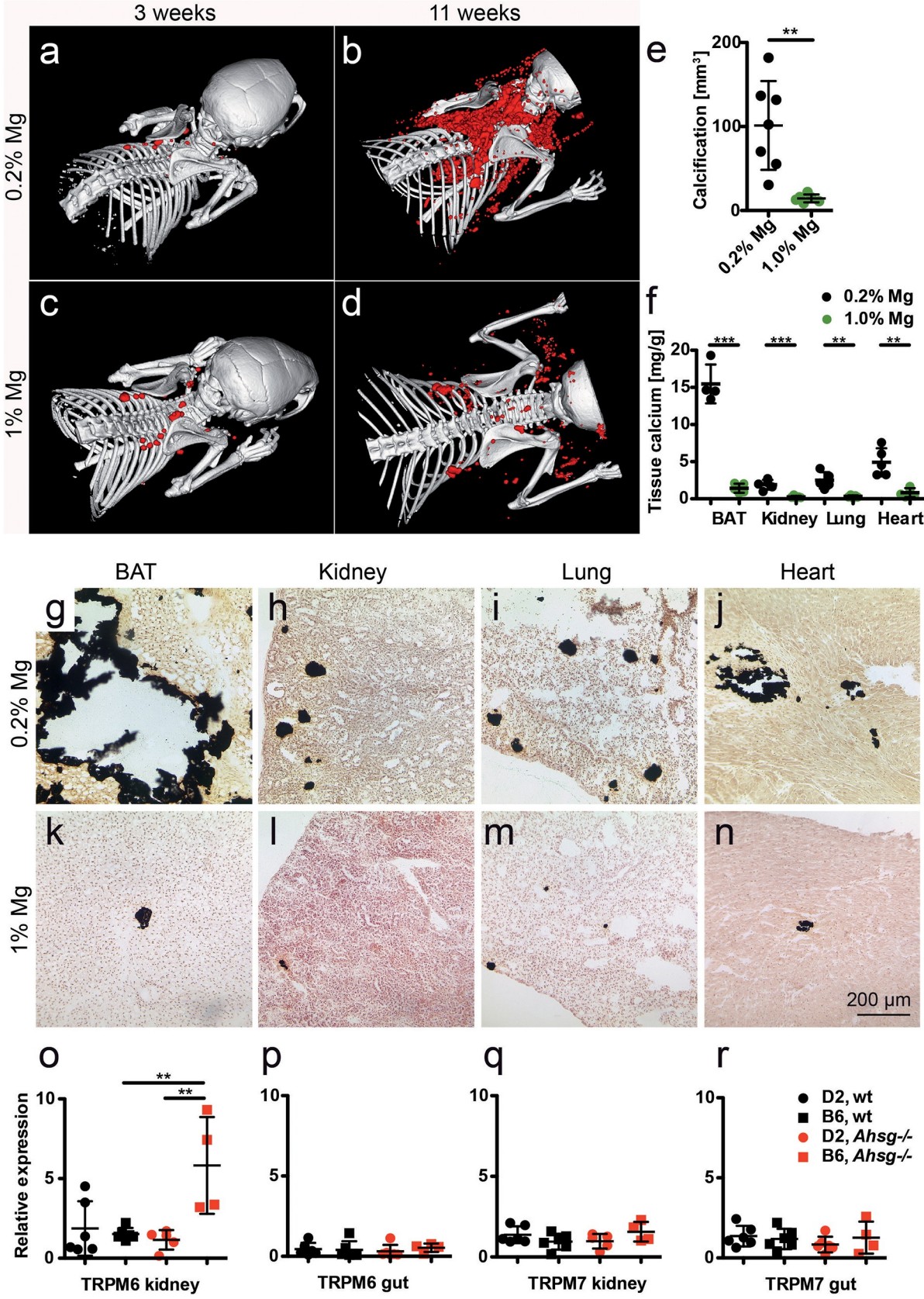

**Fig 3. Dietary magnesium supplementation attenuates soft tissue calcification in D2, *Ahsg*-/- mice. a-d**. Three-week-old mice were fed for eight weeks normal chow containing 0.2% magnesium, or a high magnesium diet containing 1% magnesium. Computed tomography of upper torso shows sparse calcified lesions in mice at the start of the feeding experiment (**a,c**), massive calcification after eight weeks on normal magnesium chow (**b**), and strongly attenuated calcification after eight weeks on high magnesium chow (**d**). **e**, Calcified tissue volumes determined at age 11 weeks by CT and segmentation. **f**, Tissue calcium content was measured following chemical extraction of brown adipose tissue (BAT), kidney, lung and heart. **g-n** show von Kossa stained cryosections of **g,k**, BAT, **h,l**, kidney, **i,m**, lung, and **j,n**, heart. Scale bar indicates 200 μm. Mice on normal chow had large calcified lesions in their BAT, occasional lesions in kidney and lung, and fibrosing calcified lesions in myocard (**b,f,g-j**). In contrast, mice fed with high magnesium developed sparse calcified lesions in the same organs (**d,f,k-n**). **o-r,** Unlike mice with the calcification resistant genetic background B6, mice with the calcification-prone genetic background D2 failed to induce Trpm6 mRNA expression in the kidney, which resulted in the functional hypomagnesemia described in Fig 1H. Student t-test (**e,f**) and one-way ANOVA with Tukey multiple comparison test (**o-r**) for statistical significance, **p<0.01, ***p<0.001.

## Dietary phosphate controls soft tissue calcification

Elevated serum FGF23 levels in DBA/2 mice compared to C57BL/6 suggested dysregulated phosphate handling despite apparently normal serum phosphate, a situation reminiscent of many CKD patients. We therefore studied the influence on the calcification phenotype of high dietary phosphate, (0.8% Pi), normal chow (0.4% Pi), and low dietary phosphate (0.2% Pi). Quantitative CT analysis illustrated in Fig 4A–4G show that during the 8-week feeding period from week 3 to 11 postnatal, mice on high phosphate developed $318 \pm 130$ mm$^3$ calcified lesions, mice on normal chow developed 3.1-fold less calcifications ($101 \pm 53$ mm$^3$, p = 0.002) and mice on low phosphate developed 1.8-fold less calcifications yet again ($55 \pm 18$ mm$^3$, p = 0.03). Fig 4H illustrates an orthogonal analytical approach measuring chemical tissue calcium in BAT, kidney, lung and heart. High phosphate feeding increased BAT calcium 2.3-fold compared to normal chow ($56.31 \pm 32.87$ mg/g vs. $24.85 \pm 13.75$ mg/g, p = 0.041), while low phosphate resembled normal chow ($31.19 \pm 16.03$ mg/g organ, p = 0.430). Similarly, high phosphate feeding increased lung calcium 2.8-fold ($8.40 \pm 2.39$ mg/g vs. $2.97 \pm 1.33$ mg/g, p = 0.0003), while low phosphate resembled normal chow ($2.48 \pm 1.38$ mg/g, p = 0.488). High phosphate feeding increased myocardial calcium 1.8-fold ($8.71 \pm 2.54$ mg/g vs. $4.81 \pm 1.55$ mg/g, p = 0.006), and low phosphate slightly decreased myocardial calcium content to 0.7-fold ($3.33 \pm 1.51$ mg/g, p = 0.084). Kidney calcium was similar in animals on normal chow ($2.98 \pm 1.98$ mg/g), on high phosphate ($1.74 \pm 0.86$ mg/g, p = 0.182) or on low phosphate diet ($1.67 \pm 0.72$ mg/g, p = 0.103). Thus CT volume analysis (Fig 4G), chemical tissue calcium analysis (Fig 4H) and von Kossa histology (Fig 4I–4T) congruently indicated that high phosphate feeding increased soft tissue calcification while low phosphate feeding decreased calcification in BAT, lung and myocard, but not in kidney.

## Parenteral pyrophosphate supplementation attenuates soft tissue calcification

Next we asked if supplementation of pyrophosphate would affect the calcification phenotype of fetuin-A deficient DBA/2 mice. Starting at postnatal week 3, we injected daily for eight weeks, i.p. boluses of 0.10 g/kg bodyweight sodium pyrophosphate. Quantitative CT analysis (Fig 5A–5C) showed that pyrophosphate injected mice developed 11-fold less interscapular calcifications within the first 11 weeks postnatal than did age-matched saline injected mice ($8.6 \pm 6.2$ mm$^3$ vs. $94.7 \pm 54.2$ mm$^3$, p = 0.002). Orthogonal methods chemical tissue calcium analysis and von Kossa histology confirmed that pyrophosphate injection decreased calcifications in all tissues analyzed (Fig 5D–5L). Parenteral pyrophosphate compared to saline reduced BAT calcium 3.1-fold ($9.93 \pm 13.62$ vs. $30.94 \pm 17.42$, p = 0.028), kidney calcium 29-fold ($0.10 \pm 0.03$ mg/g vs. $2.92 \pm 1.72$ mg/g, p = 0.002), lung calcium 14-fold ($0.26 \pm 0.074$ mg/g vs. $3.64 \pm 2.31$ mg/g, p = 0.004), and myocardial calcium 11.3-fold ($0.50 \pm 0.22$ mg/g vs. $5.63 \pm 2.90$, p = 0.0009).

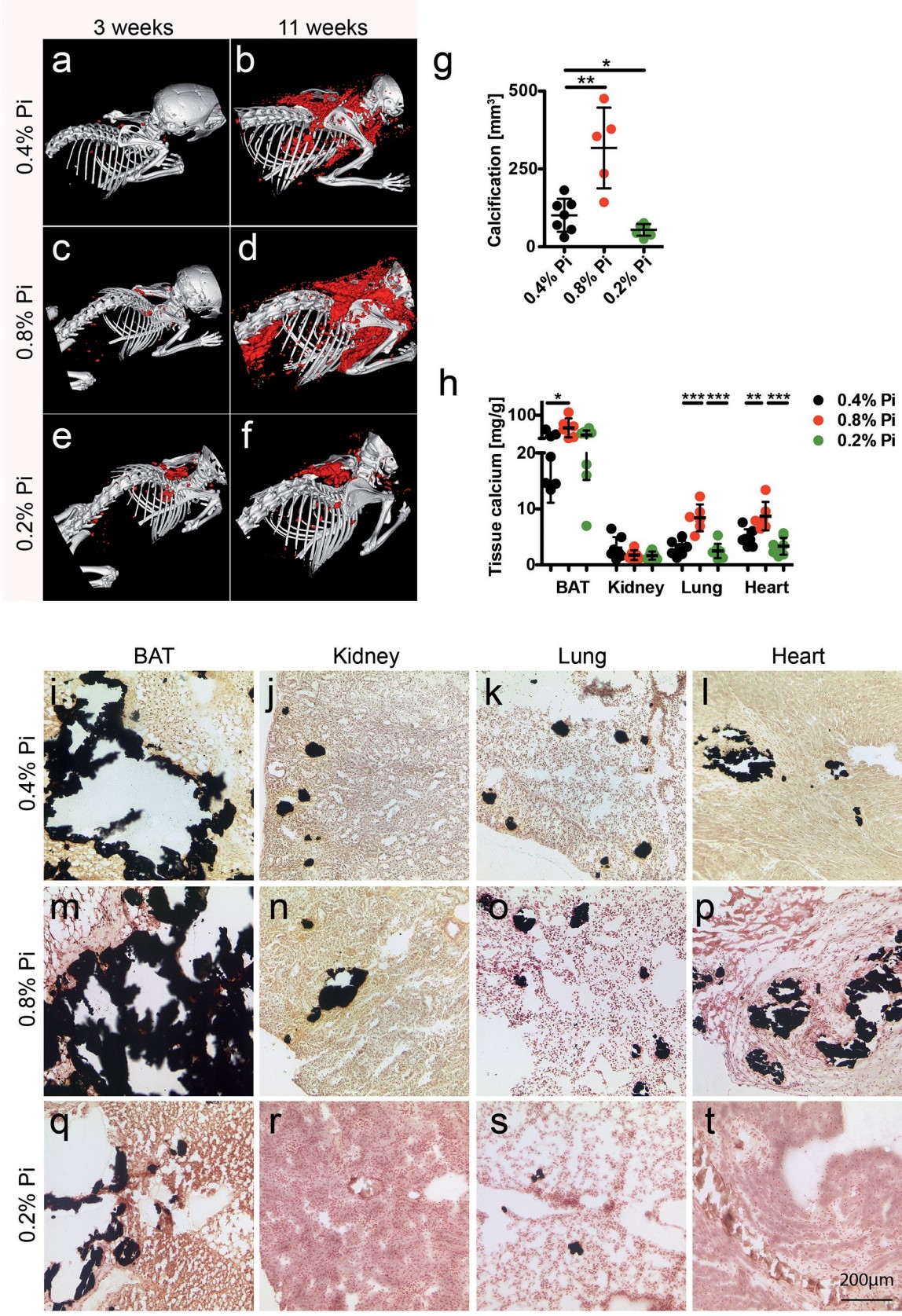

**Fig 4. Dietary phosphate controls soft tissue calcification in D2, *Ahsg-/-* mice. a-f.** Three-week-old mice were fed for eight weeks with normal chow containing 0.4% phosphorous (Pi) or with a low or high phosphate diet containing 0.2% and 0.8% phosphorous, respectively. Computed tomography of upper torso shows sparse calcified lesions in mice at the start of the feeding experiment (**a,c,e**). After feeding 0.8% Pi the animals developed markedly more calcified lesions than control animals (0.4% Pi, **b,d**), while animals on low phosphate diet (0.2%) show strongly attenuated calcification (**f**). **g,** Calcified tissue volumes determined by CT and segmentation. **h,** Tissue calcium content was measured following chemical extraction of brown adipose tissue (BAT), kidney, lung and heart. **i-t** show von Kossa stained cryosections of **i,m,q,** BAT, **j,n,r,** kidney, **k,o,s,** lung, and **l,p,t,** heart. Scale bar indicates 200 μm. **i-l,** Mice on 0.4% Pi chow had large calcified lesions in their BAT, occasional lesions in kidney and lung, and fibrosing calcified lesions in myocard. **m-p,** Mice on high 0.8% Pi diet had increased calcified lesions and **q-t,** mice on low 0.2% Pi diet had reduced calcified lesions in all organs studied. Student t-test (**g**) and one-way ANOVA with Tukey's multiple comparison test (**h**) for statistical significance, $^*p{<}0.05$, $^{**}p{<}0.01$, $^{***}p{<}0.001$.

Effectively, the calcification phenotype of pyrophosphate treated fetuin-A deficient DBA/2 mice was similar to untreated fetuin-A deficient C57BL/6 mice (Fig 1D), suggesting that pyrophosphate deficiency in DBA/2 mice contributes to soft tissue calcification.

## Reduced Abcc6 mRNA and protein expression in DBA/2 mice is caused by a hypomorphic gene mutation

Next we studied the expression of Abcc6, an ATP cassette transporter protein involved in extracellular pyrophosphate metabolism [23, 26]. The single nucleotide polymorphism (SNP) rs32756904 signifying a hypomorphic 5-bp deletion in the Abcc6 transcript [27], which was readily detected by differential PCR and DNA gel electrophoresis in all DBA/2 mice, but never in C57BL/6 mice independent of the fetuin-A genotype (Fig 6A). S3 Fig shows a representative DNA sequencing trace depicting the G>A mutation, which causes the hypomorphic SNP rs32756904. The mutation creates an additional splice donor site of exon 14 of the murine *Abcc6* gene of DBA/2 mice resulting in reduced mRNA expression. Fig 6B and 6C illustrate that kidney and liver, the two major Abcc6 producing organs indeed expressed up to 20-fold less mRNA in DBA/2 mice than in C57BL/6 mice. Two-way ANOVA of the mRNA expression data of DBA/2 mice (kidney wt 0.23±0.16, Ahsg-/- 0.15±0.08; liver wt 2±0.96, Ahsg -/- 0.49 ±0.38) vs. C57BL/6 mice (kidney wt 4.23±2.53, Ahsg-/- 2.04±0.33; liver wt 15.42±13.81, Ahsg-/- 7.06±5.07) indicated that genetic background accounted for most of the data variance in both kidney (46%, p = 0.0002) and liver (30.9%, p = 0.001). Ahsg genotype accounted for much less variance in kidney (6.8%, p = 0.08) and liver (7.5%, p = 0.17), and the interaction of background and Ahsg genotype accounted for 5.9% variance (p = 0.1) in kidney and 3.6% (p = 0.34) in liver, respectively. Together with the fact, that the SNP illustrated in Fig 6A and S3 Fig was present in all DBA/2 mice, but never in C57BL/6 mice, the expression data confirmed that the hypomorphic G>A mutation was strain specific and indeed mediated reduced Abcc6 mRNA expression. We studied by immunoblotting if reduced mRNA expression caused reduced protein expression. Fig 6D shows that the reduced liver mRNA expression indeed caused 2.4-fold reduced Abcc6 protein expression in livers of wildtype DBA/2 mice vs. C57BL/6 mice (0.42 ± 0.19 vs. 0.99 ± 0.01, p = 0.04), and 5.6-fold reduced Abcc6 protein expression in fetuin-A deficient DBA/2 mice vs. C57BL/6 mice (0.27 ± 0.03 vs. 1.42 ± 0.42, p = 0.0002). Collectively the gene expression data suggest that the hypomorphic SNP rs32756904 mutation resulted in reduced mRNA and protein expression and thus ultimately in diminished extracellular PPi.

## Hypomorphic Abcc6 mutation segregates with calcification phenotype in progeny of fetuin-A deficient DBA/2 x C57BL/6 hybrid mice

To confirm a role of Abcc6 and thus of pyrophosphate metabolism in the strong calcification phenotype of fetuin-A deficient DBA/2 mice, we analyzed by linkage analysis the contribution

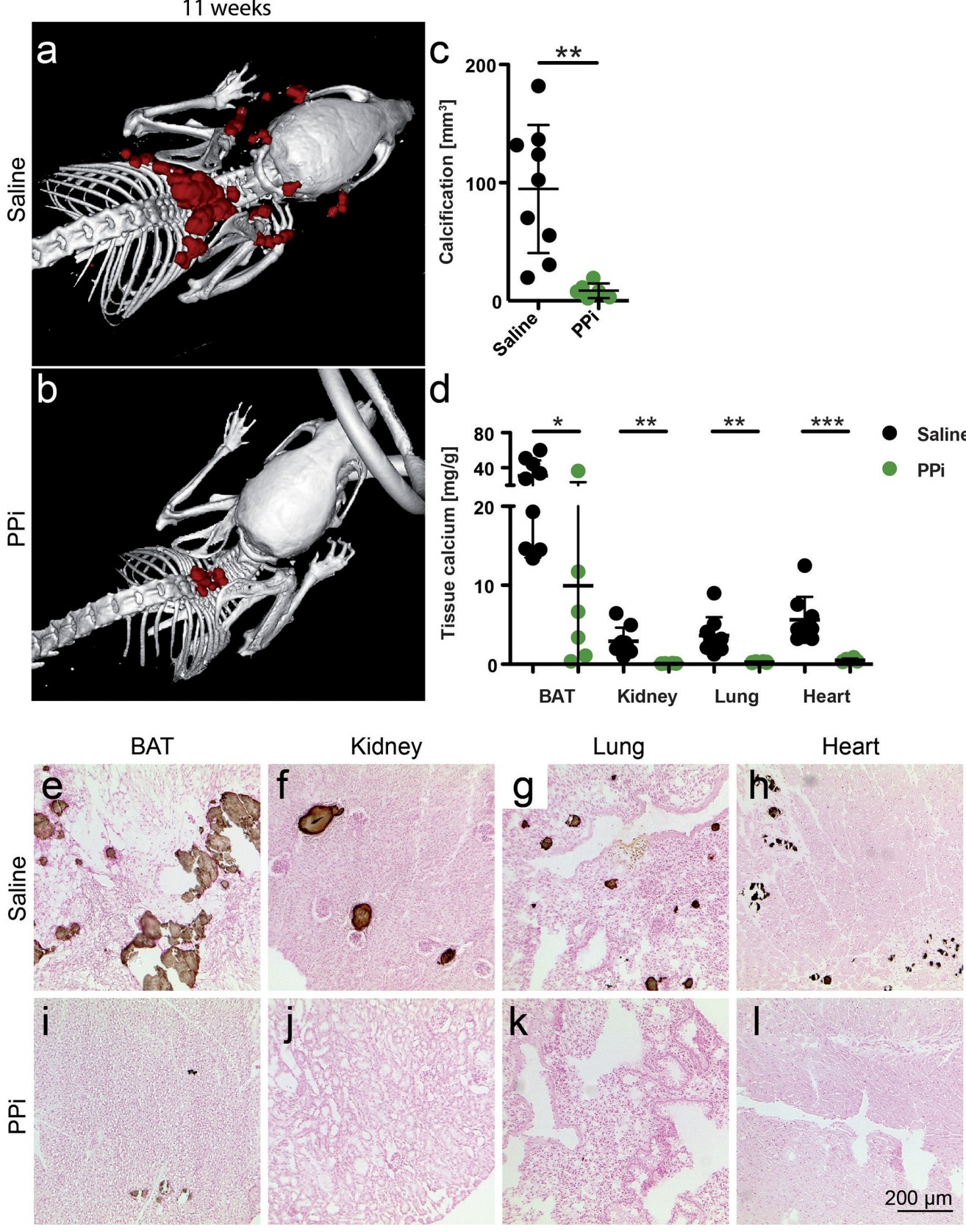

**Fig 5. Parenteral pyrophosphate supplementation attenuates soft tissue calcification in D2, *Ahsg-/-* mice. a,c,d,e-h** Three-week-old mice were injected daily for eight weeks with saline or **b-d,i-l**, with 0.10 g/kg bodyweight sodium pyrophosphate (PPi). Computed tomography of upper torso shows massive calcification after eight weeks of saline injection (**a**), and strongly attenuated calcification after eight weeks of PPi injection (**b**). **c**, Calcified tissue volumes determined by CT and segmentation. **d**, Tissue calcium content was measured following chemical extraction of brown adipose tissue (BAT), kidney, lung and heart. **e-l** show von Kossa stained cryosections of **e,i**, BAT, **f,j**, kidney, **g,k**, lung, and **h,l**, heart. Scale bar indicates 200 μm. **a,e-h**, Saline treated mice had numerous calcified lesions in their BAT, occasional lesions in kidney and lung, and fibrosing calcified lesions in myocard. **b,i-l**, In contrast, PPi treated mice had few calcified lesions only in interscapular BAT. Student t-test for statistical significance, *p<0.05, **p<0.01, ***p<0.001.

of this hypomorphic *Abcc6* mutation to the overall calcification risk. To this end we crossed fetuin-A deficient DBA/2 male with C57BL/6 female mice. The heterozygous F$_1$ offspring were randomly intercrossed to create an F$_2$ generation (Fig 7A). We analyzed calcification in 177 F$_2$ offspring (86 males, 91 females) by quantitative CT and stratified the mice with respect to calcification: **I**, no detectable calcified lesions, 65 animals; **II**, single calcified lesion in brown adipose tissue, 42 animals; **III**, two clearly separated lesions in brown adipose tissue, 42 animals; **IV**, calcified lesions in brown adipose tissue, myocardium and adrenal fat pad, 24 animals; **V**,

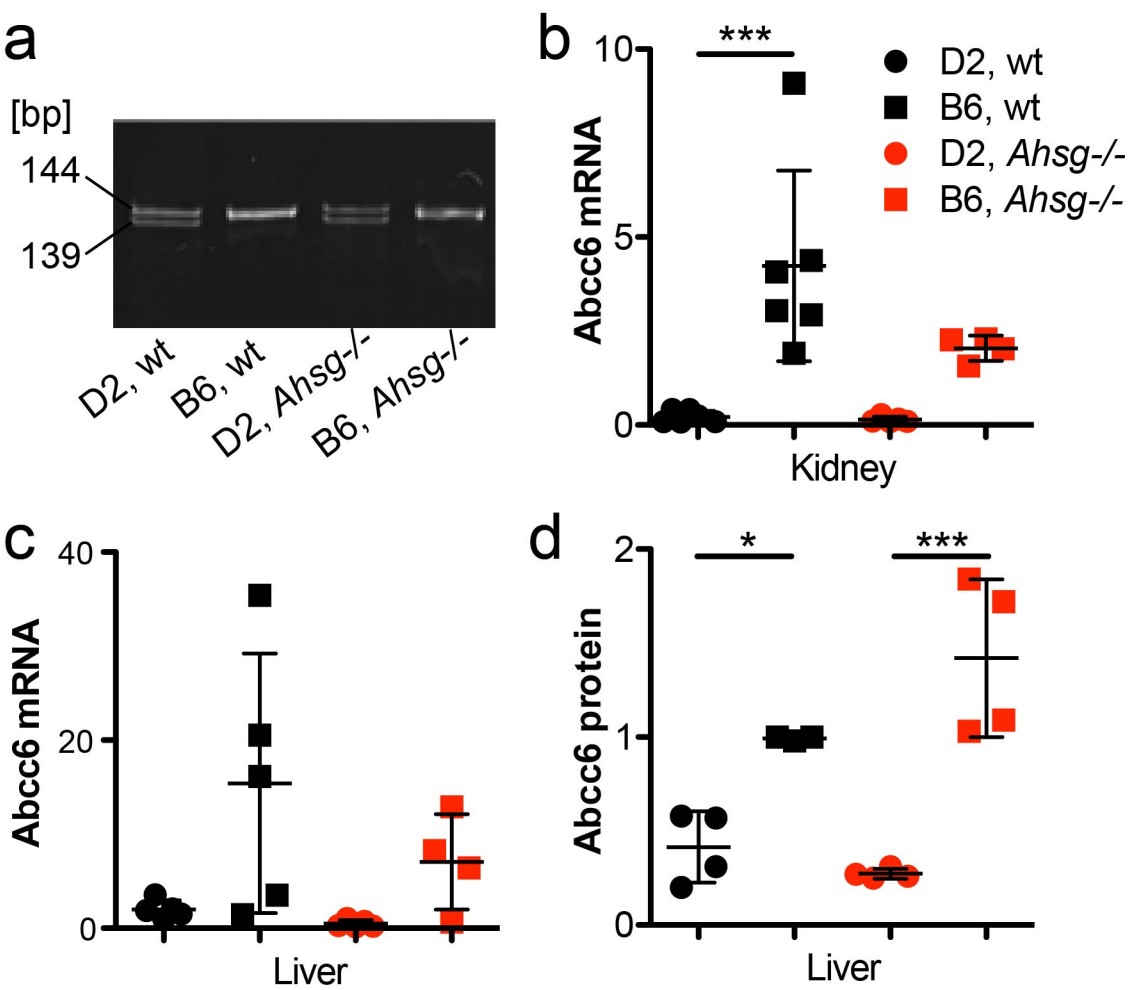

**Fig 6. Reduced Abcc6 mRNA and protein expression in DBA/2 mice is caused by a hypomorphic gene mutation. a**, SNP rs32756904 causing a 5-bp deletion in the Abcc6 transcript was detected by differential PCR and DNA gel electrophoresis. Amplicon size is given in base pairs [bp]. **b,c**, qPCR of mRNA in kidney and liver show reduced expression of correctly spliced Abcc6 mRNA in DBA/2 vs. C57BL/6 mice. **d**, Immunoblotting demonstrates correspondingly reduced Abcc6 protein expression of Abcc6 protein in the liver. Data in **b-d** are relative units. For a genomic analysis of the hypomorphic Abcc6 mutation see S3 Fig. One-way ANOVA with Tukey multiple comparison test for statistical significance, *p<0.05, ***p<0.001.

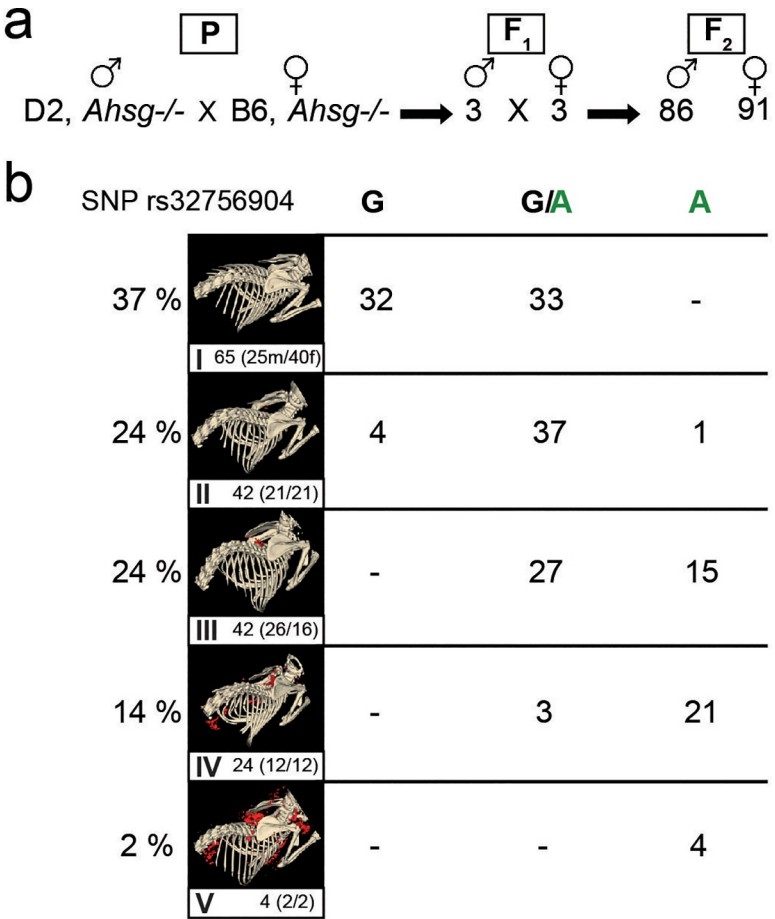

**Fig 7. Hypomorphic Abcc6 mutation segregates with calcification phenotype in progeny of fetuin-A deficient DBA/2 x C57BL/6 hybrid mice. a**, One male D2, *Ahsg-/-* mouse was naturally mated with one female B6, *Ahsg-/-* mouse (parental generation–P). **b**, Three breeding pairs of their hybrid progeny (F$_1$) produced 177 F$_2$-offspring (86 male, 91 female), which were genotyped by PCR and genomic sequencing for the hypomorphic Abcc6 single nucleotide polymorphism rs32756904. Calcification was individually scored by computed tomography and segmentation of non-skeletal mineralized tissue (red color). Calcification was scored in five categories ranging from no calcified lesions detected (**I**), one calcified lesion detected in the brown adipose tissue BAT (**II**), more than one calcified lesion detected in BAT, but not in other tissues (**III**), numerous calcified lesions detected in BAT, kidney and myocard (**IV**), and numerous massive calcified lesions in BAT, kidney, myocard and axillary skin (**V**). The clear segregation of scores with SNP rs32756904 suggests that at the most 1–2 additional unlinked genes besides Fetua/Ahsg, Trpm6, and Abcc6 determine the severity of the calcification phenotype of the mice.

calcified lesions throughout the body, 4 animals (Fig 7B). Heterozygous F$_1$ progeny had at most one calcified lesion in brown adipose tissue, indicating that half the gene dosage of calcification regulatory genes from resistant C57BL/6 genetic background was sufficient to prevent widespread calcification even in the absence of fetuin-A. Fig 7B illustrates that F$_2$ mice carrying homozygous the hypomorphic mutant A allele of *Abcc6* had severe calcifications (III, IV, V; with one exception in category II). Mice with the homozygous wildtype G allele had hardly any detectable calcified lesions (I, II). Mice carrying both the mutant and the wildtype allele (G/A) had variable calcification including categories I, II, III, IV. This result suggested a necessary, but not an exclusive determination by the SNP rs32756904 of the strong calcification phenotype of fetuin-A deficient mice. Presently we cannot offer a similar frequency plot regarding *Trpm6* genetics, but the clear segregation of the mice with little or no calcification (I, II) and the mice with severe calcification (III, IV, V) with no overlap suggested that at the most 1–2

additional unlinked genes beyond the genes studied here, namely *Fetua*/*Ahsg*, *Trpm6*, and *Abcc6* determined the severity of the calcification propensity of the mice.

## Discussion

Mineralization regulators evolved to prevent crystallization and deposition of calcium mineral in tissues not meant to mineralize. Fetuin-A is a potent systemic regulator of ectopic calcification mediating the solubilization and clearance of circulating protein-mineral complexes, which might otherwise cause calcification [28, 29]. In this present study parenteral supplementation of fetuin-A inhibited the formation of ectopic calcifications in a fetuin-A deficient mouse model. In humans, complete fetuin-A deficiency was not reported until recently despite the fact that human fetuin-A/a2HS glycoprotein was a commonly used paternity marker [30]. Partial fetuin-A deficiency was, however, reported in many studies of CKD, and correlated with increased serum calcification propensity [31–33]. Recently, genetic fetuin-A deficiency was found associated with infantile cortical hyperostosis (Caffey disease) [34], a pathology compatible with the post-weaning epiphysiolysis causing distal femur dysplasia and foreshortened hindlimbs in fetuin-A-deficient mice maintained against the genetic background C57BL/6 [35]. Unlike the fetuin-A deficient mice maintained against the background DBA/2 studied here, these mice had no overt calcification outside the skeleton, suggesting that growth plate cartilage, a tissue physiologically meant to calcify during bone development, readily calcifies, when fetuin-A is lacking, while other tissues require additional calcification risk factors. Indeed, this is the main notion of this current work.

We showed that DBA/2 mice had functional hypomagnesemia regardless of their fetuin-A genotype due to altered expression of magnesium transporter Trpm6, which mediates intestinal absorption and kidney reabsorption of magnesium, respectively. The protective action of Mg in calcification is known [36, 37], as is its basic regulation [24]. The adaptive regulation of Mg in calcification was previously observed in adenine-treated rats [38] and it is known that fetuin-A and magnesium can substitute each other in reducing calcification propensity [39]. We hypothesized that DBA/2 mice are more prone to calcification than are C57BL/6 mice, because of renal Mg wasting [40] and thus loss of systemic Mg, a potent crystal nucleation inhibitor [41]. A recent study showed that kidney-specific Trpm6 knockout mice had no phenotype in contrast to intestine-specific knockout [42]. These results recapitulated clinical studies with patients with Trpm6 null mutations clearly showing that a defect in intestinal $Mg^{2+}$ uptake causes hypomagnesemia. In any case, hypomagnesemia compounds the deficiency of serum fetuin-A, a crystal growth inhibitor. Like in this study, magnesium supplementation prevented calcification in a mouse model of pseudoxanthoma elasticum (PXE), *Abcc6*-/- mice [13], and reduced nephrocalcinosis [43] as well as vascular and soft tissue calcifications in a rat CKD model [44]. Serum Mg is considered a poor marker of Mg deficiency and is not routinely monitored in CKD patients [37]. However, in CKD patients, treatment with Mg containing phosphate binders, supplementation with Mg-oxide or increased Mg in dialysate, all reduced serum calcification propensity [45] and/or the progression of cardiovascular calcifications [46]. These findings are in line with our observation that magnesium supplementation prevented ectopic calcification in DBA/2 mice. Hypomagnesemia in humans is common in about 15% of the general population. Inherited hypomagnesemia is rare [47] but malnutrition and acquired metabolic disease seem to account for most cases. Drugs such as proton pump inhibitors also reduce serum magnesium [48, 49]. When hypomagnesemia combines with calcification risk factors including CKD, it might accelerate the development of ectopic calcification. However, in CKD patients, Mg supplementation must be tightly monitored to avoid hypermagnesemia, despite reports of it being safe [45].

DBA/2 mice regardless of their fetuin-A genotype had hypomagnesemia associated with elevated serum FGF23. Likewise, rats on magnesium-deficient diet had elevated serum FGF23 [50] and CKD cats had inverse association of serum Mg and FGF23 independent of serum phosphate [51]. Collectively these results suggest that serum FGF23 may be associated with hypomagnesemia in addition to the well-established association with hyperphosphatemia. CKD-associated hyperphosphatemia is a result of impaired renal phosphate excretion and is directly associated with cardiovascular calcification [52, 53]. Additionally, elevated phosphate levels might promote coronary atheroma burden in non-uremic patients [54]. Thus, dietary phosphate restriction is paramount in dialysis patient care [55]. Although several studies report on the effectiveness of phosphate binders on calcification inhibition in different animal studies [56–58], the effect of dietary phosphate restriction on calcification has been studied in surprisingly little detail [59, 60]. Dietary phosphate restriction in CKD patients remains challenging, because it generally implies protein restriction and thus potential malnutrition.

Dystrophic calcification in genetically predisposed mouse strains such as C3H/He, DBA/2, BALB/c and 129S1 is a complex genetic trait still not fully understood. Genetic mapping identified a single major locus, designated *Dyscalc1*, located on proximal chromosome 7, and three additional modifier loci associated with dystrophic cardiac calcinosis (DCC) following myocardial injury [61]. Integrative genomics identified the gene mapping to the *Dyscalc1* locus as *Abcc6*, the ATP-binding cassette C6 [27]. Abcc6 is a membrane transporter with ambiguous substrate specificity. Forced expression of Abcc6 was associated with increased cellular release of ATP, which rapidly breaks down into AMP and the potent calcification inhibitor pyrophosphate PPi [23, 26, 62]. PPi is a calcium phosphate crystal nucleation and growth inhibitor that interferes with crystal lattice formation. Circulating PPi is mostly liver-derived and relatively long-lived [23]. In hemodialysis patients, low PPi levels [63] are associated with increased calcification risk [64]. We found reduced plasma PPi levels in D2 mice compared to B6 mice, which resulted from reduced Abcc6 expression in D2 mice. Recent studies showed that plasma PPi deficiency is critically involved in dystrophic calcification in *Abcc6-/-* mice [65]. A noncoding splice variant of Abcc6 was deemed responsible for the PPi deficiency in calcification prone C3H/He mice. A single nucleotide polymorphism rs32756904 encodes an additional splice donor site and thus the hypomorphic splice variant of Abcc6. We detected rs32756904 in all of our DBA/2 mice, but never in C57BL/6 mice underscoring the critical contribution of genetic background to ectopic calcification [66, 67]. Reduced Abcc6 protein expression was also detected in BAT mitochondria of DBA/2J vs. C57BL/6J mice [68] extending our results shown in Fig 6 for kidney and liver Abcc6 expression. At the same time this finding suggests a mechanistic explanation of the calcification proneness of BAT observed in this study, namely lack of PPi in a metabolically highly active tissue. Metabolically active tissue are by necessity rich in mitochondria, which have long been known as preferred starting sites of dystrophic calcification in e.g. myocardium [69], a tissue heavily calcified and compromised also in our mice (Figs 2–5 and [19]).

Intriguingly, a recent systems genetics study conducted in a large panel of isogenic but diverse strains of mice originating from a C57BL/6 x DBA/2 intercross showed that the most variable trait was alkaline phosphatase (ALPL), varying 6.5-fold between 44 and 287 U/L [70]. Increased ALPL in DBA/2 mice likely further reduces by increased lysis the levels of PPi, which are already diminished due to low Abcc6 expression. ALPL is thus a strong candidate to fill the 1–2 gene gap suggested by our Mendelian segregation study of the Abcc6 hypomorh (Fig 7) to fully account for the soft tissue calcification of fetuin-A deficient DBA/2 mice. Hypomorphic mutations in the *Abcc6* gene were deemed responsible for both dystrophic calcification in mice and for PXE in humans [27]. Children suffering from generalized arterial calcification of infancy (GACI), caused by a mutation of the gene *Enpp1*, which is responsible

for the conversion of ATP into AMP and PPi, could also benefit from PPi supplementation therapy. In addition to PXE and GACI, Hutchinson-Gilford progeria syndrome (HGPS), was linked to reduced PPi levels associated with the development of extracellular matrix calcification [12]. Although mouse models for HGPS, GACI and PXE show promising results when treated with PPi, human studies remain to be conducted. Human and animal intervention studies have recently become much more feasible, because—contrary to long held belief, unknowingly also shared by us—PPi can simply be supplemented orally [71].

The orthogonal study of PPi-treated fetuin-A deficient DBA/2 mice by quantitative CT, von Kossa histology and chemical tissue calcium analysis returned similar, but by no means identical representations of soft tissue calcification. For instance, treatment reduced the volume of tangible calcified lesions in BAT 11-fold in CT, yet reduced tissue calcium only 3-fold (Fig 5A–5E and 5I). Along these lines high dietary phosphate increased peripheral soft tissue calcification but hardly affected the kidney, a principal organ involved in phosphate homeostasis (Fig 4). These findings indicate that saturating mineral loading of the ECM precedes calcified lesion formation, and that low flux peripheral organs are more prone to ECM calcification than e.g. kidney, a high flux excretory organ. Further, PPi may inhibit triggering events of initial lesion formation and growth. ECM mineral loading is not routinely studied in soft tissue calcification, but may well be an important feature of chronic mineral imbalance. Precedent may be taken from experimental salt-sensitive hypertension, were high-salt diet in rats led to interstitial hypertonic $Na^+$ accumulation in skin, recruiting macrophages that ultimately regulated salt-dependent volume and blood pressure [72]. Similarly, chronic mineral loading of ECM with calcium, phosphate and uremic toxins may ultimately trigger immune cell-dependent "inflammaging" thus driving e.g. arterial calcification in children on dialysis [73].

In conclusion, we describe that a compound triple deficiency of fetuin-A, magnesium and pyrophosphate in DBA/2 fetuin-A knockout mice leads to arguably one of the most severe soft tissue calcification phenotypes known. We show that parenteral fetuin-A, dietary magnesium, low dietary phosphate, and parenteral pyrophosphate supplementation all prevented the formation of calcified lesions in fetuin-A deficient DBA/2 mice suggesting that similar therapeutic approaches might also reduce early stage CKD-associated cardiovascular calcifications and calciphylaxis. Notably, these calcifications occurred without any signs of osteogenic conversion of calcifying cells, underscoring the importance of extracellular mineral balance to prevent unwanted mineralization [20].

## Supporting information

**S1 Fig. Serum immunoblot confirms complete lack of fetuin-A expression (*Ahsg*) in D2, *Ahsg-/-* and B6, *Ahsg-/-* mice.**
(JPG)

**S2 Fig. Fetuin-A supplementation attenuates soft tissue calcification in D2, *Ahsg-/-* mice. a-d**, Three-week-old mice were injected i.p. five times a week for three weeks, with saline (**a,c**) or with 0.24 g/kg bodyweight fetuin-A (**b,d**). Lateral and dorsal radiographs show brown adipose tissue and renal fat calcification (arrows) in saline treated but not in fetuin-A treated mice.
(JPG)

**S3 Fig. Genomic organization of SNP rs32756904 in the gene *Abcc6* on chromosome 7 in D2 and B6 fetuin-A (*Ahsg*) deficient mice.** B6 mice have unambiguous splice donor and acceptor sites in their genomic DNA (gDNA) resulting in a single pre-mRNA and a single mRNA transcript. In contrast, D2 mice carry a G>A mutation in their gDNA, creating an

additional splice donor site five base pairs upstream of the wildtype splice donor site. This creates an alternative five base pairs shortened splice variant of the mRNA transcript, which is not translated. Thus, the hypomorphic SNP rs32756904 results in reduced expression of functional Abcc6 protein (Fig 6D) and reduced extracellular pyrophosphate levels (Fig 1I).
(JPG)

**S1 Table. Primer sequences used for quantitative real-time PCR.**
(DOCX)

## Acknowledgments

This work was supported by grants awarded to WJD by the IZKF Aachen of the Medical Faculty of RWTH Aachen and by the German Research Foundation (DFG SFB/TRR219-Project C-03). We thank Koen van de Wetering (Thomas Jefferson University, Philadelphia, USA) for help with the pyrophosphate assay.

## Author Contributions

**Conceptualization:** Willi Jahnen-Dechent.

**Data curation:** Willi Jahnen-Dechent.

**Funding acquisition:** Willi Jahnen-Dechent.

**Investigation:** Anne Babler, Carlo Schmitz, Andrea Buescher, Marietta Herrmann, Willi Jahnen-Dechent.

**Methodology:** Felix Gremse, Theo Gorgels.

**Resources:** Willi Jahnen-Dechent.

**Writing – original draft:** Anne Babler, Carlo Schmitz, Andrea Buescher, Willi Jahnen-Dechent.

**Writing – review & editing:** Anne Babler, Carlo Schmitz, Andrea Buescher, Marietta Herrmann, Theo Gorgels, Juergen Floege, Willi Jahnen-Dechent.

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
