## [Decision Letter · Decision Letter 0]

4 Dec 2019

PONE-D-19-27520

Microvasculopathy And Soft Tissue Calcification In Mice Are Governed by Fetuin-A, Pyrophosphate And Magnesium

PLOS ONE

Dear Prof Jahnen-Dechent,

Thank you for submitting your manuscript to PLOS ONE. After careful consideration, we feel that it has merit but does not fully meet PLOS ONE’s publication criteria as it currently stands. Therefore, we invite you to submit a revised version of the manuscript that addresses the points raised during the review process.

We would appreciate receiving your revised manuscript by Jan 18 2020 11:59PM. To enhance the reproducibility of your results, we recommend that if applicable you deposit your laboratory protocols in protocols.io, where a protocol can be assigned its own identifier (DOI) such that it can be cited independently in the future. For instructions see: http://journals.plos.org/plosone/s/submission-guidelines#loc-laboratory-protocols

We look forward to receiving your revised manuscript.

Kind regards,

Elena Aikawa, MD PhD

Academic Editor

PLOS ONE

Journal Requirements:

All animal experiments were conducted in agreement with German animal protection law and were approved by the state animal welfare committee.

Additional Editor Comments (if provided):

Reviewers' comments:

Reviewer's Responses to Questions

**Comments to the Author**

1. Is the manuscript technically sound, and do the data support the conclusions?

Reviewer #1: Partly

Reviewer #2: Yes

2. Has the statistical analysis been performed appropriately and rigorously? 

Reviewer #1: Yes

Reviewer #2: Yes

3. Have the authors made all data underlying the findings in their manuscript fully available?

Reviewer #1: Yes

Reviewer #2: Yes

4. Is the manuscript presented in an intelligible fashion and written in standard English?

Reviewer #1: Yes

Reviewer #2: Yes

5. Review Comments to the Author

Reviewer #1: This study demonstrates differences in ectopic calcification between DBA/2 and C57BL/6 strains deficient in the mineralization inhibitor, fetuin-A. The different responses in the strains is attributed to decreased levels of other calcification inhibitors in the serum. The authors also note a SNP that affects expression of Abcc6, a regulator of pyrophosphate, may also explain the difference between the mouse strains. Overall, the study is interesting, well-written, and timely. The premise of the study is also strong in that the phenotypic differences between the mouse strains is striking and does not seem to involve differences in cellular osteogenesis. The comments below may help strengthen the conclusions drawn in the study.

1. p. 4: Has it been previously shown that Mg compensates for fetuin-A loss, or is this a new finding?

2. p. 4 (and other points in the text): The authors should provide more data in the text. For example, the results text focused on Fig. 2 describe differences between groups, but the data shown in the graphs often do not seem to indicate statistical significance. It would be helpful if the authors could provide means, standard deviations, and absolute p values (i.e., not < or >) to help increase the rigor of the conclusions made.

3. pp. 4-5: How does the serum Mg levels differ in the mice fed the Mg supplement? Does the serum Mg reach levels high enough to explain the differences in C57BL/6 and DBA/2 phenotypes?

4. p. 5: How did the dietary phosphate affect circulating phosphate levels in the mice? The observations that calcification changes based upon differences in dietary phosphate are striking, but does altered phosphate handling tested through these experiments necessarily explain the difference between the mouse strains? Would similar changes be observed by placing the C57BL/6 mice on diets with different dietary phosphate? If the C57BL/6 mice show the same rate of change (i.e., the slope of the line that compares dietary phosphate to calcification size is the same between mouse strains), this would indicate that the baseline calcification of the DBA/2 mouse is higher than the C57BL/6 mouse, but that it is not due to altered phosphate handling. If this is the case, it could mean that the alterations in mineral inhibitors are more important in explaining the phenotypes.

5. p. 5: The data on PPi supplementation do not necessarily lead to the conclusion that PPi deficiency is the major determinant of dystrophic calcification in DBA/2 mice. In fact, the data indicate that circulating PPi isn't different between the mouse strains. Adding more PPi, a known mineral inhibitor, does inhibit the mineral formation, but this could just mean that elevated PPi compensates for inhibitors that are different (e.g., Mg).

6. pp. 5-6: Given the noted changes in Abcc6, can the authors comment why serum PPi does not seem to be decreased in the DBA/2 mice?

7. p. 6: Did the authors try to relate the level of Abcc6 (e.g., by western blotting) to the measured calcification in each mouse (or at least a subset of mice from each category)? This could help strengthen the conclusion that altered Abcc6 is responsible for the phenotypic differences observed.

Reviewer #2: General comments:

Mice lacking fetuin-A suffer from severe ectopic calcification in the DBA/2 background, whereas those in the C57BL/6 background barely develop calcification. By analyzing these fetuin-A deficient mice with different genetic backgrounds, the authors identified pyrophosphate and magnesium as the major determinants of the difference in calcification propensity. The data were convincing and well presented. This reviewer has only a few minor concerns.

Minor points:

If the authors still have serum samples remaining, they might want to measure serum magnesium levels (Figure 3) to see if high magnesium diet increased serum magnesium in D2 Ahsg-/- mice to the level equivalent to that in B6 Ahsg-/- mice.

The authors might want to add some explanation to Discussion on general information on TRPM6 function in magnesium handling, e.g., TRPM6 in the intestine and kidney mediates absorption and reabsorption of magnesium, respectively.

Where in the kidney and lung was calcification observed, e.g. renal tubules, glomerulus, interstitial space, blood vessels etc.?

Page 5, line 29: Fig 4d-l -> Fig 5d-l

Page 9, line 13: enclosed -> enrolled

6. PLOS authors have the option to publish the peer review history of their article (what does this mean?). If published, this will include your full peer review and any attached files.

Reviewer #1: No

Reviewer #2: Yes: Makoto Kuro-o

---

## [Author Response · Author response to Decision Letter 0]

15 Jan 2020

Response To Reviewers

We would like to thank both reviewers for their constructive criticism, their time and effort to improve the quality of our manuscript. We respond to their comments point-by-point. To improve reading we thoroughly edited the manuscript for clarity and consistency. We added references to the discussion to address the points raised by the reviewers.

Reviewer #1:

1. p. 4: Has it been previously shown that Mg compensates for fetuin-A loss, or is this a new finding?

Our answer: We and others previously showed that Mg independently reduces calcification propensity. 

Braake, ter, A.D., Eelderink, C., Zeper, L.W., Pasch, A., Bakker, S.J.L., De Borst, M.H., Hoenderop, J.G.J., and de Baaij, J.H.F. (2019). Calciprotein particle inhibition explains magnesium-mediated protection against vascular calcification. Nephrol Dial Transpl 2, 1241–1249.

Braake, ter, A.D., Shanahan, C.M., and de Baaij, J.H.F. (2017). Magnesium Counteracts Vascular Calcification: Passive Interference or Active Modulation? Arterioscler Thromb Vasc Biol 37, 1431–1445.

Braake, ter, A.D., Tinnemans, P.T., Shanahan, C.M., Hoenderop, J.G.J., and de Baaij, J.H.F. (2018). Magnesium prevents vascular calcification in vitro by inhibition of hydroxyapatite crystal formation. Sci Rep 8, 2069.

A study in adenine induced calcification in rats showed that reduced serum fetuin-A was associated with increased serum Mg (https://doi.org/10.1038/ki.2008.700).

We added the latter two references to the discussion as ref 38 and 39.

“The adaptive regulation of Mg in calcification was previously observed in adenine-treated rats [38] and it is known that fetuin-A and magnesium can substitute each other in reducing calcification propensity [39].

38. Matsui I, Hamano T, Mikami S, Fujii N, Takabatake Y, Nagasawa Y, et al. Fully phosphorylated fetuin-A forms a mineral complex in the serum of rats with adenine-induced renal failure. Kidney Int. 2009;75(9):915-28. doi: 10.1038/ki.2008.700. PubMed PMID: 19190677.

39. ter Braake AD, Eelderink C, Zeper LW, Pasch A, Bakker SJL, De Borst MH, et al. Calciprotein particle inhibition explains magnesium-mediated protection against vascular calcification. Nephrol Dial Transplant. 2019;2:1241-9. doi: 10.1093/ndt/gfz190.”

2. p. 4 (and other points in the text): The authors should provide more data in the text. For example, the results text focused on Fig. 2 describe differences between groups, but the data shown in the graphs often do not seem to indicate statistical significance. It would be helpful if the authors could provide means, standard deviations, and absolute p values (i.e., not < or >) to help increase the rigor of the conclusions made.

Our answer: We thank the reviewer for this very constructive criticism. We added the data throughout the results section as requested. We uploaded unedited photographs documenting the Abcc6 Western Blots in a file called “Western blot Documentation” We are prepared to also provide the original Prism files on calcification quantitation by CT and by chemical analysis on request. When we re-examined the quantitative data, we noted an interesting discrepancy in measuring calcification by quantitative CT, and the orthogonal chemical analysis of tissue calcium. The fold-differences detected by CT volumetric imaging were often greater than the differences in chemical analysis. We attribute this to the different threshold sensitivities of the methods. CT imaging can only detect dense calcified lesions, while chemical calcium analysis will detect any degree of ECM mineral loading. We included this observation and our conclusion at the end of the discussion.

“The orthogonal study of PPi-treated fetuin-A deficient DBA/2 mice by quantitative CT, von Kossa histology and chemical tissue calcium analysis returned similar, but by no means identical representations of soft tissue calcification. For instance, treatment reduced the volume of tangible calcified lesions in BAT 11-fold in CT, yet reduced tissue calcium only 3-fold (Figs 5a-e, i). Along these lines high dietary phosphate increased peripheral soft tissue calcification but hardly affected the kidney, a principal organ involved in phosphate homeostasis (Fig 4). These findings indicate that saturating mineral loading of the ECM precedes calcified lesion formation, and that low flux peripheral organs are more prone to ECM calcification than e.g. kidney, a high flux excretory organ. Further, PPi may inhibit triggering events of initial lesion formation and growth. ECM mineral loading is not routinely studied in soft tissue calcification, but may well be an important feature of chronic mineral imbalance. Precedent may be taken from experimental salt-sensitive hypertension, were high-salt diet in rats led to interstitial hypertonic Na+ accumulation in skin, recruiting macrophages that ultimately regulated salt-dependent volume and blood pressure [72]. Similarly, chronic mineral loading of ECM with calcium, phosphate and uremic toxins may ultimately trigger immune cell-dependent “inflammaging” thus driving e.g. arterial calcification in children on dialysis [73].”

3. pp. 4-5: How does the serum Mg levels differ in the mice fed the Mg supplement? Does the serum Mg reach levels high enough to explain the differences in C57BL/6 and DBA/2 phenotypes?

Our answer: We did not analyze serum Mg in the supplemented mice, but focused on CT imaging and histology of calcification as an endpoint. The mice had soft stools, a hallmark sign of elevated Mg intake. We relied on work by van den Broek and Beynen who triggered dystrophic calcification in DBA/2 wildtype mice with low dietary magnesium (https://doi.org/10.1258/002367798780599758). We figured that high dietary magnesium should diminish calcification in fetuin-A deficient DBA/2 mice, which it did. We added the reference as ref 36.

“The protective action of Mg in calcification is known [36, 37], as is its basic regulation [24]. 

36. van den Broek FA, Beynen AC. The influence of dietary phosphorus and magnesium concentrations on the calcium content of heart and kidneys of DBA/2 and NMRI mice. Laboratory animals. 1998;32(4):483-91. PubMed PMID: 9807763.”

4. p. 5: How did the dietary phosphate affect circulating phosphate levels in the mice? The observations that calcification changes based upon differences in dietary phosphate are striking, but does altered phosphate handling tested through these experiments necessarily explain the difference between the mouse strains? Would similar changes be observed by placing the C57BL/6 mice on diets with different dietary phosphate? If the C57BL/6 mice show the same rate of change (i.e., the slope of the line that compares dietary phosphate to calcification size is the same between mouse strains), this would indicate that the baseline calcification of the DBA/2 mouse is higher than the C57BL/6 mouse, but that it is not due to altered phosphate handling. If this is the case, it could mean that the alterations in mineral inhibitors are more important in explaining the phenotypes.

Our answer: High dietary phosphate increased and low dietary phosphate decreased calcification in this and many more studies from our and other labs. We did in fact show this earlier for C57BL/6 mice in conjunction with nephrectomy (Westenfeld NDT 2007) (https://doi.org/10.1258/002367798780599758). There is no strict correlation between plasma phosphate and calcification, probably because the excess phosphate gets deposited in calcified lesions. Please also see our discussion about ECM mineral loading above. 

5. p. 5: The data on PPi supplementation do not necessarily lead to the conclusion that PPi deficiency is the major determinant of dystrophic calcification in DBA/2 mice. In fact, the data indicate that circulating PPi isn't different between the mouse strains. Adding more PPi, a known mineral inhibitor, does inhibit the mineral formation, but this could just mean that elevated PPi compensates for inhibitors that are different (e.g., Mg).

Our answer: We respectfully disagree and maintain that the levels of PPi are different within the confines of methodology (see next point). This is not surprising given the reduced expression of Abcc6 mRNA and protein. We also would like to point out that parenteral PPi supplementation reduced calcification as did dietary magnesium and parenteral fetuin-A. This is the most significant finding of the entire study and indeed the title of the paper. We fully agree with reviewer that one may substitute for the other in stabilizing the fluid phase and thus aiding removal of potentially calcifying entities. Therefor we happily tone down any statement of PPi being “the major determinant of calcification in DBA/2 mice”. We now close the results on PPi supplementation (page 18) by stating 

“Effectively, the calcification phenotype of pyrophosphate treated fetuin-A deficient DBA/2 mice was similar to untreated fetuin-A deficient C57BL/6 mice (Fig 1d), suggesting that pyrophosphate deficiency in DBA/2 mice contributes to soft tissue calcification.”

6. pp. 5-6: Given the noted changes in Abcc6, can the authors comment why serum PPi does not seem to be decreased in the DBA/2 mice?

Our answer: Plasma PPi is notoriously difficult to measure, especially in the small amounts of plasma that can be retrieved from mice. We sought help from Koen de Wetering, an expert in PPi and we mention this in acknowledgement. To measure PPi, plasma must be platelet-free, which is achieved by centrifuge filter devices, which hardly pass enough liquid for the PPi measurements. Small amounts of lysed platelets will drastically increase the PPi readings. We can only point to similar studies by others in the field and to the unambiguous beneficial effects of PPi supplementation. The pertinent results section now reads:.

“Plasma levels of inorganic pyrophosphate (PPi), another systemic calcification inhibitor, were elevated in C57BL/6 mice compared to DBA/2 mice independent of fetuin-A genotype (2.68 ± 1.23 µM vs. 1.87 ± 0.99 µM in wildtype mice, p=0.75 and 3.27 ± 1.46 µM vs. 1.52 ± 0.94 µM in fetuin-A deficient mice, p=0.18, Fig 1i). Measured differences were not statistically significant due to large standard deviation, a known complication of measuring PPi in small volumes of plasma.”

7. p. 6: Did the authors try to relate the level of Abcc6 (e.g., by western blotting) to the measured calcification in each mouse (or at least a subset of mice from each category)? This could help strengthen the conclusion that altered Abcc6 is responsible for the phenotypic differences observed.

Our answer: We did in fact confirm reduced Abcc6 protein expression in the parent animals (Fig. 6b). We include the Western blot for the reviewers’ discretion. In these mice, the amount expressed was tightly associated with the SNP genotype, which we analyzed in all F2 mice. We also cited recent literature showing reduced Abcc6 expression in mitochondria of DBA/2 mice vs. C57BL/6 mice and increased phosphatase expression in DBA/2 mice vs. C57BL/6, both of which reduce PPi. (new refs 68-69)

“We detected rs32756904 in all of our DBA/2 mice, but never in C57BL/6 mice underscoring the critical contribution of genetic background to ectopic calcification [66, 67]. Reduced Abcc6 protein expression was also detected in BAT mitochondria of DBA/2J vs. C57BL/6J mice [68] extending our results shown in Fig 6 for kidney and liver Abcc6 expression. At the same time this finding suggests a mechanistic explanation of the calcification proneness of BAT observed in this study, namely lack of PPi in a metabolically highly active tissue. Metabolically active tissue are by necessity rich in mitochondria, which have long been known as preferred starting sites of dystrophic calcification in e.g. myocardium [69], a tissue heavily calcified and compromised also in our mice (Figs 2-5 and [19]).

Intriguingly, a recent systems genetics study conducted in a large panel of isogenic but diverse strains of mice originating from a C57BL/6 x DBA/2 intercross showed that the most variable trait was alkaline phosphatase (ALPL), varying 6.5-fold between 44 and 287 U/L [70]. Increased ALPL in DBA/2 mice likely further reduces by increased lysis the levels of PPi, which are already diminished due to low Abcc6 expression. ALPL is thus a strong candidate to fill the 1-2 gene gap suggested by our Mendelian segregation study of the Abcc6 hypomorh (Fig 7) to fully account for the soft tissue calcification of fetuin-A deficient DBA/2 mice.”

Reviewer #2 comments:

If the authors still have serum samples remaining, they might want to measure serum magnesium levels (Figure 3) to see if high magnesium diet increased serum magnesium in D2 Ahsg-/- mice to the level equivalent to that in B6 Ahsg-/- mice.

Our answer: Unfortunately, no serum samples are left to analyze magnesium levels of the mice. As discussed above, we did not analyze serum Mg in these mice because we focused on imaging of calcification as an endpoint. However, the mice had soft stools, a hallmark sign of elevated Mg intake.

The authors might want to add some explanation to Discussion on general information on TRPM6 function in magnesium handling, e.g., TRPM6 in the intestine and kidney mediates absorption and reabsorption of magnesium, respectively.

Our answer: We added a short discussion as suggested.

Where in the kidney and lung was calcification observed, e.g. renal tubules, glomerulus, interstitial space, blood vessels etc.?

Our answer: We report a related study “Lumenal Calcification and Microvasculopathy in Fetuin-A-Deficient Mice Lead to Multiple Organ Morbidity” (ref. 20), which studied the earliest calcifications. In this study we conclude, that a calcium phosphate precipitates as a protein-rich coagulum in the lumen of the microvasculature and large lesions rapidly replace the surrounding tissue (Herrmann et al., PLOS One, under review)

Page 5, line 29: Fig 4d-l -> Fig 5d-l

Page 9, line 13: enclosed -> enrolled

Thank you for spotting these errors. Corrected!

---

## [Decision Letter · Decision Letter 1]

28 Jan 2020

Microvasculopathy and soft tissue calcification in mice are governed by fetuin-A, magnesium and pyrophosphate

PONE-D-19-27520R1

Dear Dr. Jahnen-Dechent,

We are pleased to inform you that your manuscript has been judged scientifically suitable for publication and will be formally accepted for publication once it complies with all outstanding technical requirements.

With kind regards,

Elena Aikawa, MD PhD

Academic Editor

PLOS ONE

Additional Editor Comments (optional):

Reviewers' comments:

Reviewer's Responses to Questions

**Comments to the Author**

1. If the authors have adequately addressed your comments raised in a previous round of review and you feel that this manuscript is now acceptable for publication, you may indicate that here to bypass the “Comments to the Author” section, enter your conflict of interest statement in the “Confidential to Editor” section, and submit your "Accept" recommendation.

Reviewer #1: All comments have been addressed

Reviewer #2: All comments have been addressed

2. Is the manuscript technically sound, and do the data support the conclusions?

Reviewer #1: (No Response)

Reviewer #2: Yes

3. Has the statistical analysis been performed appropriately and rigorously? 

Reviewer #1: (No Response)

Reviewer #2: Yes

4. Have the authors made all data underlying the findings in their manuscript fully available?

Reviewer #1: (No Response)

Reviewer #2: Yes

5. Is the manuscript presented in an intelligible fashion and written in standard English?

Reviewer #1: (No Response)

Reviewer #2: Yes

6. Review Comments to the Author

Reviewer #1: (No Response)

Reviewer #2: (No Response)

7. PLOS authors have the option to publish the peer review history of their article (what does this mean?). If published, this will include your full peer review and any attached files.

Reviewer #1: No

Reviewer #2: Yes: Makoto Kuro-o

---

## [Editor Report · Acceptance letter]

4 Feb 2020

PONE-D-19-27520R1 

Microvasculopathy and soft tissue calcification in mice are governed by fetuin-A, magnesium and pyrophosphate 

Dear Dr. Jahnen-Dechent:

I am pleased to inform you that your manuscript has been deemed suitable for publication in PLOS ONE. Congratulations! Your manuscript is now with our production department. 

With kind regards,

on behalf of

Dr. Elena Aikawa 

Academic Editor

PLOS ONE